# Characteristics of Mercury Pollution and Ecological Risk Assessment in Different Degraded Grasslands of the Songnen Plains, Northeastern China

Zhaojun Wang [1,2,†], Lei Wang [1,2,†], Gang Zhang [1,2,3,4,*], Xu Li [1], Xiangyun Li [1], Yangjie Zhang [1], Xuhang Zhou [1], Ming Chen [1], Tingting Xiao [1], Zhili Feng [1], Yue Weng [1], Zhanhui Tang [1,2,*] and Deli Wang [1,2,3,4]

1   School of Environment, Northeast Normal University, Changchun 130117, China; wangzj217@nenu.edu.cn (Z.W.); wangl788@nenu.edu.cn (L.W.); lix896@nenu.edu.cn (X.L.); lixy869@nenu.edu.cn (X.L.); zhangyj415@nenu.edu.cn (Y.Z.); zhouxh561@nenu.edu.cn (X.Z.); chenm593@nenu.edu.cn (M.C.); xiaott070@nenu.edu.cn (T.X.); fengzl306@nenu.edu.cn (Z.F.); w124493606@163.com (Y.W.); z35976113@163.com (D.W.)
2   State Environmental Protection Key Laboratory of Wetland Ecology and Vegetation Restoration, Changchun 130117, China
3   Key Laboratory of Vegetation Ecology, Ministry of Education, Northeast Normal University, Changchun 130117, China
4   Institute of Grassland Science, Northeast Normal University, Changchun 130022, China
*   Correspondence: zhangg217@nenu.edu.cn (G.Z.); tangzh789@nenu.edu.cn (Z.T.)
†   Z.W. and L.W. are co-first authors of the article.

**Abstract:** Mercury (Hg) is a global and widely distributed heavy metal pollutant. Mercury can affect human health as well as the health of ecosystems and poses ecological risks. The subjects of this study are three types of grassland in the Beidianzi region, Songnen Plains, Northeastern China, characterized by different degrees of degradation. The mercury content levels in the atmosphere, soil, and forage grass on the different grasslands were determined. In addition, the relationships between the mercury pollution levels in the atmosphere and soil, and the mercury distribution correlations between the soil and plants, were examined in detail. The potential risk index (RI), single factor index (PI), and ground accumulation index (Igeo) were used to evaluate the ecological risks. The results showed that the mercury content in the soils of three types of grassland exceeded the China national standard (GB36600-2018), and the soil mercury content in the moderately degraded grassland was the highest. The single factor index method and land accumulation index method showed that the three types of grassland were slightly polluted, while the potential risk index showed that the three types of grassland were severely polluted, and the potential risk index of the moderately degraded grassland was the highest. The potential risk index decreased with the increase of soil depth. The variation trend of atmospheric mercury content was lower in the morning and evening and higher in the afternoon. The potential risk index of atmospheric mercury indicated that all types of grassland were at severe risk. There was a significant positive correlation between atmospheric mercury and soil mercury. The mercury content in herbage increased with the increase of degradation. The BP neural network prediction model constructed had good accuracy and had certain reference value.

**Keywords:** mercury; Songnen Plains; atmospheric; soil; degradation; risk assessments

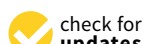



## 1. Introduction

Mercury is a toxic heavy metal widely distributed in the environment. It has high persistence, is widely distributed in ecosystems, and is known to have biological enrichment abilities in the food chain. Mercury has major impacts on human health and ecological environments [1–5]. Especially for humans, indirect or direct exposure to mercury can cause serious harm [6–8]. Therefore, mercury pollution and the ecological risks caused by its high toxicity and its widespread existence have attracted considerable attention in

recent years. At the present time, the accumulation levels of mercury in soil around the world are very high. The content levels of mercury in the soil range between 250 and 1000 Gg [9]. Natural sources of mercury include soil geology, forest fires, and volcanic eruptions [10]. However, a large proportion of mercury is attributed to anthropogenic influences [11]. Mercury in the atmosphere mainly enters soil surfaces through dry and wet deposition, and the soil surfaces release mercury through complex physicochemical and biochemical reactions [12]. Through these various chemical reactions, the mercury entering the soil will migrate, aggregate, and disperse between the soil's internal and external environment. Furthermore, temperature factors may also lead to the release of large amounts of mercury trapped in the soil [13]. Therefore, soil–atmosphere interface fluxes are important components of global and regional mercury biogeochemical cycles [14–16].

It is also well known that the migration and release processes of mercury are influenced by a variety of factors. For example, the factors affecting the release of mercury from soil include the types and concentration levels of mercury in the soil, along with atmospheric mercury concentration levels [17,18]. In addition, such meteorological factors as atmospheric pressure, temperature, wind speeds and turbulence, solar radiation, and snow cover may have impacts on the aforementioned processes [14,19]. It has also been found that soil water content [20], soil temperatures [21], etc., may be other influencing factors. Previous studies regarding mercury have mainly focused on the migration and transformation of mercury in various ecosystems and the ecological risks they present [14,22–24]. In addition, risk assessments for different types of mercury have been completed in order to facilitate long-term management processes.

The risks of the heavy metal contamination of soil have received increased attention in recent years [25]. Moreover, with the further development of various industries, heavy metal mercury emissions continue to increase. For example, in the Victoria Goldfield Lake region of Tanzania (Africa), approximately 3 to 4 tons of mercury have been determined to be released into the atmosphere each year [26]. On a global scale, significant increases in mercury content will undoubtedly lead to potential ecological risks [27]. With the widespread use of pollution risk assessments, researchers have also introduced risk assessment models into the study of mercury pollution in grassland and forage areas, which include ecological environment and human health dimensions. At the present time, the most commonly used assessment method is the potential ecological risk assessment index. However, in order to comprehensively reflect the pollution situation of mercury in grassland regions and its migration mechanism between atmosphere, soil, and forage, more than three methods are generally used for the risk evaluations [24].

Grassland ecosystems account for approximately 30% of the world's land area [28,29]. Those areas provide many important ecosystem service values, such as C and N storage, primary production, ecosystem diversity maintenance, water and soil conservation, and cultural services. During the past several decades, with the further development of urbanization, the mercury pollution levels in air and soil environments have become increasingly serious, which has attracted worldwide attention [30]. As a transfer station between soil and the atmosphere, plants play an important role in the cycle of mercury. Due to the particularity of mercury, it can be enriched in plants through the atmosphere and soil. For example, atmospheric mercury can fall into the surface wax layer of the upper and lower leaves of plants through dry and wet deposition, and plants can absorb mercury in the atmosphere while opening stomata during photosynthesis and respiration [31,32]. In addition, plant roots can also absorb mercury in soil [33]. At the present time, it is considered that the absorption of mercury by plants is still mainly through dry and wet deposition from the atmosphere and soil [34]. Mercury enrichment in vegetation can also be a source of atmospheric mercury [12,35,36]. Mercury pollution in grasslands mainly accumulates in grassland surface ecosystems through dry and wet deposition [37]. Therefore, the process of plants absorbing atmospheric mercury and soil mercury in grasslands is very important in the global mercury cycle.

The Changling-Songyuan area of the Jilin Province is located in an ecotone of agriculture and animal husbandry, with abundant grass resources. However, at the present time, few risk assessments have been conducted regarding grassland herbage. Therefore, in the current investigation, based on the special properties of mercury and the general background of increasingly serious mercury pollution, the ecological risks brought about by the migration of mercury from the atmosphere to the soil and from the soil to plants were examined. This study is of great significance for understanding mercury pollution in pasture areas and the rational management and layout of grasslands. The main discussion points of this study included the following points and issues:

1.  The background values and dynamics of mercury content in soil, vegetation, and atmosphere driven by different degrees of soil degradation in degraded grassland ecosystems, which represent the distribution differences;
2.  The correlations of the mercury between the soil and herbage and the atmosphere and soil, and the possible sources and influencing factors of the mercury;
3.  An evaluation index system was established. A single factor pollution index method and a ground accumulation index method were used to evaluate the mercury pollution status of different grassland types. Then, the degrees of pollution of the different grassland types were compared in order to determine the controlling factors of the spatial distributions of mercury in the soil and atmosphere of different degraded grassland types;
4.  The potential ecological risks of mercury pollution in grassland regions with different degrees of degradation, within the Songnen Plain region, were comprehensively assessed.

## 2. Materials and Methods

### 2.1. Description of the Study Area

This study's sampling area was located near Yaojingzi Ranch in Changling County, which is situated in the western section of Jilin Province, Northeastern China (Figure 1), between 123°6′–124°45′ E and 43°59′–44°42′ N. The area is rich in pasture, but it is also a serious salinization problem area. Due to the local geomorphology, the rainwater cannot flow out after the soluble carbonate deposits, which then accumulates in the region. There is little human activity near the study area, and there is no large-scale mining and industrial production. The main sources of mercury in this grassland are atmospheric mercury deposition and background mercury in soil.

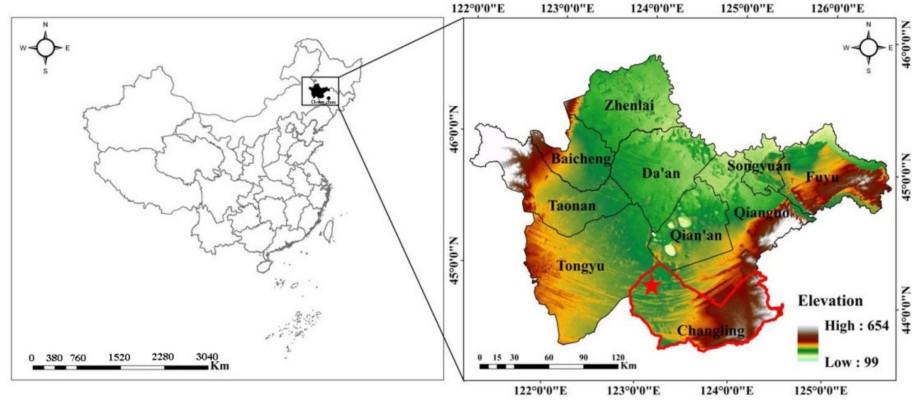

**Figure 1.** Location of the study area.

The study area is characterized by a semi-arid and sub-humid temperate monsoon climate. The spring seasons are short, dry, and windy. The summer seasons are warm and rainy. The temperature differences between the day- and night-time hours were found to be the greatest during the autumn months. Winter in the study area is generally long and cold. The climate varies widely in the region, with the month of January being extremely

cold and August being hot, and the rainfall is mainly concentrated during the months from June to August (Table 1).

**Table 1.** Study regional climate conditions.

| Project | Values |
|---|---|
| The average duration of sunshine | 2880 h |
| The annual solar radiation | 125 kcal/cm$^2$ |
| The annual average number of windy days | more than 100 days |
| The annual average wind speed ranged | 4–5 m/s |
| The annual average rainfall ranged | 400–500 mRn |
| The minimum rainfall value | 250 mm |
| The annual evaporation rate | 1500 and 2000 mm |

### 2.2. Different Degradation Degrees of the Grassland Types

Changling County is a typical meadow grassland area located in the Songnen Plains. The vegetation is mainly composed of six species families, including *Compositae*, *Leguminosae*, and grasses. *Leymus chinensis* is widely distributed in the area and is the representative of azonal salinized meadow vegetation. According to the degree of grassland degradation (GB 19377-2003) (Table 2), the area can be divided into severely degraded grassland, moderately degraded grassland, and non-degraded grassland.

**Table 2.** Classification of grassland degradation.

| Monitoring Project | Classification of Grassland Degradation | | | |
|---|---|---|---|---|
| | Non-Degraded | Lightly Degraded | Moderately Degraded | Severely Degraded |
| Percentage reduction in total coverage (%) | 0~10 | 11~20 | 21~30 | >30 |

The severely degraded grassland area was dominated by halophyte community species, such as *Suaeda glauca* and *Salinella glauca*. *S. glauca* is an annual herb with erect stems and is considered to be a typical halophyte with high humidity, salt, alkali, and infertility tolerance. *Puccinellia distans* are perennial herbs that can tolerate alkaline soil at pH 10 or above. The aforementioned are also tolerant of seasonal water changes and can form communities of a single species, and large alkali spots were observed in the area, which indicated serious salinization issues. The soil is dense and soil bulk density is high. Small pores and poor ventilation and water permeability were also observed. The poor soil quality restricted plant growth and development, as shown in Figure 2a.

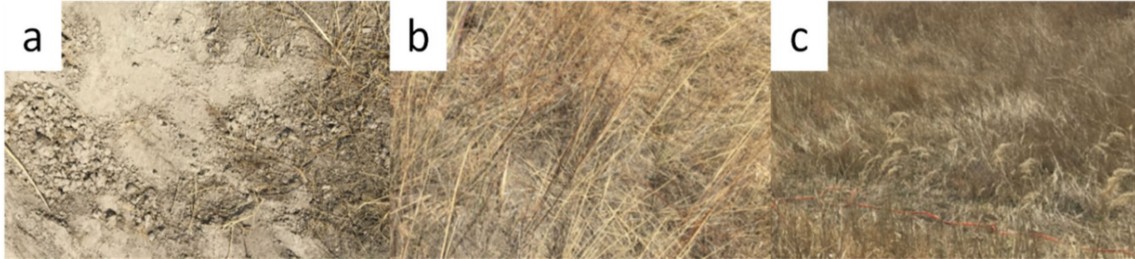

**Figure 2.** Grassland areas with different degrees of degradation: (**a**). severely degraded grassland; (**b**). moderately degraded grassland; (**c**). non-degraded grassland.

In the current investigation, the moderately degraded grassland community is dominated by miscellaneous grass community. Its manifestation as an alkaline spot is not obvious, and weeds clung to the surface of the soil. The grass layers of the fine herbage were significantly decreased, with relative increases in inferior forage or poisonous and

harmful plants. The forage production in the medium degraded grassland areas was reduced by 30% to 40% when compared with the non-degraded grassland area. The community is mainly dominated by *Artemisia alkali*, *Herba virginis*, and *L. chinensis* (Figure 2b).

The non-degraded grassland area was composed of *L. chinensis* as the building species, and the grass coverage was approximately 75%. The growth rates of the *L. chinensis* were good, and the physical and chemical properties of the soil were observed to be more suitable for plant growth. For example, the soil structure was loose and the soil bulk density was found to be small. The pores were large, and the ventilation and water permeability were beneficial for the growth of forage (Figure 2c).

### 2.3. Arrangement of the Sampling Points

In this study, three sample plots with different degradation types were selected, which represented severely degraded grassland, moderately degraded grassland, and non-degraded grassland, respectively. An area of 100 square meters was randomly assigned to each type of plot as a test plot, and a suitable sample site was randomly selected in each sample square for the purpose of conducting three measurements regarding atmospheric mercury concentrations and mercury exchange fluxes. The experimental areas measured 25 cm $\times$ 40 cm = 0.1 m$^2$. Three soil collections were made in each sample site.

### 2.4. Sample Collection and Analysis

This study's sampling process was carried out between 20 February 2019 and 10 May 2019. During the experiment, we randomly monitored the atmospheric mercury concentration for 24 h for three days, every five minutes, and recorded environmental factors, including atmospheric temperature, soil temperature, and wind speed. A soil section of 0–80 cm deep was dug out at each sample point; the soil samples were taken every 5 cm and then quickly transported to laboratory facilities in order to measure soil properties and mercury content levels. The grass samples (including upright dead bodies and litter) and the soil samples were collected simultaneously. The grass samples from the sample fields were mainly collected from the *L. chinensis* species. After collection, the samples were placed into plastic bags, labeled, and shipped back to laboratory for analysis.

An RA-915+ mercury analyzer was used to measure the concentration levels (values) of atmospheric mercury in the sampling sites (the instrument was calibrated before use). The obtained soil samples were naturally air-dried for 1 week in laboratory and then sieved through an 80-mesh sieve. The mercury content in the soil was measured using RA-915+ mercury analyzer after sieving. After the collected plant samples were brought back to the laboratory, their leaves, stems, and roots were washed with deionized water. Then, the samples were naturally dried. The biomass weight values of the roots, stems, and leaves of the plants were measured, and the mercury content levels of the roots and aboveground parts of the *Leymus* plant sample were measured using an RA-915+ mercury analyzer [38].

### 2.5. Quality Control

We analyzed all samples after quality control, which included soil (GSS5, 290 mg/kg) and liquid (GSB04- 1729-2004, 1 µ/L) reference materials (http://www.crmrm.com/, China. 16 August 2021). Primary calibration was performed at the onset of analysis and every 2–3 months thereafter or additionally as needed. The validity of the primary calibration was determined by analyzing the standards run with each batch of 48 samples. If the standards were found to be within 7.5% of their certified values, the primary calibration was assumed to be valid. Otherwise, a new primary calibration curve was established. The detection limits of this method were 0.5 µg/kg (soil) and 2 µg/kg (plant). Prior to sample analysis, we carried out tests to verify whether the Hg concentrations could be detected or if they were lower than the detection limit. We included approximately one duplicate sample with each batch of 3 samples. The relative deviation of the duplicate sample was less than 10% [38].

*2.6. Simulations of the Mercury Content Levels of the Leymus Chinensis in the Different Types Degraded Grassland Using Neural Networks*

The spatial distribution of mercury in the soil of the different types of grassland has variability. Therefore, it was difficult to construct accurate distribution maps of polluted areas and heavy metal Hg concentration using conventional methods (such as GIS). Therefore, in order to explore the factors influencing the spatial distribution of mercury content in different types of grassland containing *L. leymus*, this study adopted an intelligent estimation method of heavy metal concentrations based on artificial neural networks [39]. A maximum method and a minimum method were first adopted. The PREMNMX function in MATLAB was adopted to normalize the data. The normalization formula was as follows:

$$[x_k] = 2 \times \frac{x_k - x_{min}}{x_{max} - x_{min}} - 1 \tag{1}$$

where $[x_k]$ represents the normalized value, $X_k$ is the input vector value, and $X_{max}$ and $X_{min}$ indicate the maximum and minimum values of the input vector, respectively. Then, in order to improve the training effects and the fitting accuracy of the model, the outliers in the original data were eliminated. That is to say, the points greater than 2 times the standard deviation of the sample mean, or less than 2 times the standard deviation of the sample mean, were removed. Therefore, the sample points were guaranteed to be within the range of $[\mu - 2\sigma, \mu + 2\sigma]$.

The main goal of this study was to explore the influencing factors of elemental mercury distribution in grassland areas. However, when a neural network model's input values are too great, the fitting of the data set will be low. Therefore, we selected mercury content in soil and mercury content in atmosphere as two influencing factors, so as to explore their influence on the five parts of the upper leaves, lower leaves, upper stems, lower stems, and the roots of the sheep grass. Influencing factors with correlation coefficients greater than 0.4 (for example, the mercury content in the soil, mercury content in the atmosphere, and the wind speeds) were selected as the input data, making the number of input layer nodes 2. The output of the system included the Hg content of the upper leaves, lower leaves, upper stems, lower stems, and the roots of the *L. chinensis*. Since the five different parts were fitted separately, the node number of the output layer was 5.

In regard to the function between the input and output of the nodes of the hidden layer and the output layer, we chose an S-type function with high accuracy and low error as the transfer function, in which a linear purelin function was used for the neuron of the output layer (Formula (1)), and a sigmoid-type tansig function was used for both the neuron of the input layer and the hidden layer (Formula (2)):

$$f(x) = \frac{1}{1 + e^{-x}} \tag{2}$$

$$f(x) = \frac{1 - e^{-x}}{1 + e^{x}} \tag{3}$$

As for the training method of the neural network, since the traditional gradient descend method adjusts the weight and threshold of a network along the negative gradient direction of the network's performance parameters, the convergence speed is slow, and the memory consumption is large. Therefore, a fast convergence speed was chosen in this study, and the calculation amount was reduced due to the avoidance of the direct calculation of Hesse matrix. However, the Levenberg–Marquardt method, which requires a large amount of memory, was used as the training method in this study. Its training process is detailed in Figure 3.

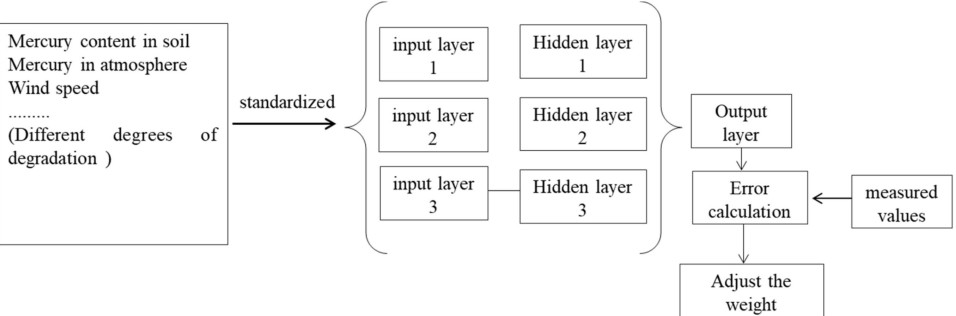

**Figure 3.** Structural chart of a BP neural network for mercury.

*2.7. Data Analysis Method*

In this study, SPSS, MATLAB, and Origin software were utilized for the statistical analyses of the data for the purpose of exploring the characteristics of the mercury fluxes between the soil and air interfaces, linear content relationships between the soil and the plants, and the influencing effects of environmental factors on the soil–air mercury fluxes in the grasslands with different degradation degrees. SPSS and other software were used to analyze the obtained data.

## 3. Results

*3.1. Analysis of the Degrees of Mercury Pollution in the Soil of Different Degraded Grasslands and Potential Risk Assessments*

### 3.1.1. Soil Mercury Content Levels of the Different Types of Degraded Grassland

The mercury content levels in the soil of the three grassland types are shown in Figure 4. The average mercury content in the non-degraded grassland was 5.84 $\mu g \cdot kg^{-1}$, with a maximum value of 11.67 $\mu g \cdot kg^{-1}$ and a minimum value of 2.43 $\mu g \cdot kg^{-1}$. The content level of mercury in the moderately degraded land was determined to be 6.96 $\mu g \cdot kg^{-1}$, with a maximum value of 11.67 $\mu g \cdot kg^{-1}$ and a minimum value of 4.90 $\mu g \cdot kg^{-1}$. Furthermore, the content level of mercury in the severely degraded land was 6.25 $\mu g \cdot kg^{-1}$, with a maximum value of 11.00 $\mu g \cdot kg^{-1}$ and a minimum value of 3.96 $\mu g \cdot kg^{-1}$. The variation coefficients of the moderately degraded grassland were greater than the non-degraded grassland and the severely degraded grassland, which indicated that in terms of spatial distribution, the mercury content distribution in soil of the moderately degraded grassland displayed the largest differences. However, the mercury content in the soil of the severely degraded grassland was lower than those of the other two grasslands. Importantly, when compared with the new national soil standards issued by China in 2018 (GB 36600-2018, national standard is 3.40 $\mu g \cdot kg^{-1}$, selecting the soil standards with pH ranging from 7.8 to 8.5), it appears that the mercury content levels in the different types of grassland all exceeded the standards, with the moderately degraded grasslands exceeding the standards by 3.4 times.

### 3.1.2. Ecological Risk Assessments of the Mercury Levels in Soil of the Different Degraded Grassland Areas

(1)  Geo-accumulation index method and single factor pollution index method

A single factor pollution index (PI) and a land accumulation index (Igeo) were used to evaluate the risk values of the soil of three different degraded grasslands. The pollution level is evaluated according to the evaluation level in Table 3. The evaluation results are detailed in Table 4. The single factor index value of the non-degraded land was 1.71, which corresponds to a slight pollution level, and the land accumulation index value was 0.19, a slightly polluted degree. The single factor index value of the moderately degraded soil was 2.03, which was also at the level of slight pollution; the land accumulation index of the moderately degraded grassland was 0.44, which indicates slight pollution. The severe degradation of the soil single factor index value was 1.83, a slight pollution level, and the cumulative index was 0.28, belonging to slightly polluted. The evaluation results of all the

grassland soils revealed that mercury pollution of the moderately degraded soil represents the highest risk. The non-degraded grassland soil had the lowest risk index for mercury pollution.

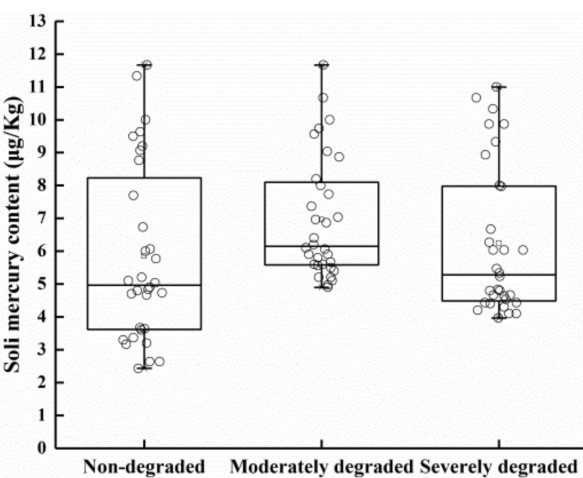

**Figure 4.** Soil mercury content in three types of grassland.

**Table 3.** Classification of the Igeo, Er, and PI.

| Type | Range | Level | Type | Range | Level | Type | Range | Level |
|------|-------|-------|------|-------|-------|------|-------|-------|
| $I_{geo}$ | $I_{geo} < 0$ | Unpolluted | Er | | | PI | | |
| | $0 \leq I_{geo} < 1$ | Unpolluted to moderately polluted | | $Er < 40$ | Low | | $PI < 0$ | Unpolluted |
| | $1 \leq I_{geo} < 2$ | Moderately polluted | | $40 \leq Er < 80$ | Moderate | | $PI \leq 1$ | Unpolluted |
| | $2 \leq I_{geo} < 3$ | Moderately to heavily polluted | | $80 \leq Er < 160$ | High | | $1 < PI \leq 2$ | Slightly polluted |
| | $3 \leq I_{geo} < 4$ | Heavily polluted | | $160 \leq Er < 320$ | Serious | | $2 < PI \leq 3$ | Moderately polluted |
| | $4 \leq I_{geo} < 5$ | Heavily to extremely polluted | | $Er \geq 320$ | Severe | | $PI \geq 3$ | Heavily polluted |
| | $I_{geo} \geq 5$ | Extremely polluted | | | | | | |

**Table 4.** Evaluation results of the total pollution in grassland areas with different degrees of degradation.

| Methods | Non-Degraded | Moderately Degraded | Severely Degraded |
|---------|--------------|---------------------|-------------------|
| PI | 1.71 | 2.03 | 1.83 |
| Igeo | 0.19 | 0.44 | 0.28 |

(2)   Potential risk evaluation index method

This study used Origin 8.0 to construct three-dimensional surface maps of three soil ecological risk indices of different degradation levels, in order to represent the spatial distributions of ecological risk indices. In addition, contour maps of atmospheric mercury ecological risk indices of the three different degraded grasslands were also drawn. The results are shown in Figure 5. Among the three types of grassland, it was found that the moderate degradation grassland had the highest potential risk index for mercury in soil. It can be seen that with the increases in the soil depth, the potential risks were reduced. Moreover, the potential risk values of mercury in the non-degraded soil were relatively low, and the potential risk values decreased with the increases in depth. However, the potential risk values of the soil with severe degradation were lower than the moderate risk values and were considered indicative of a slightly polluted level. Once again, with

the increases in sampling depth, the potential risk values decreased. The decrease of the potential risk with depth corresponds to a decrease in the mercury content. This would support the idea that the main source of mercury in the area is the long-distance transport of airborne mercury.

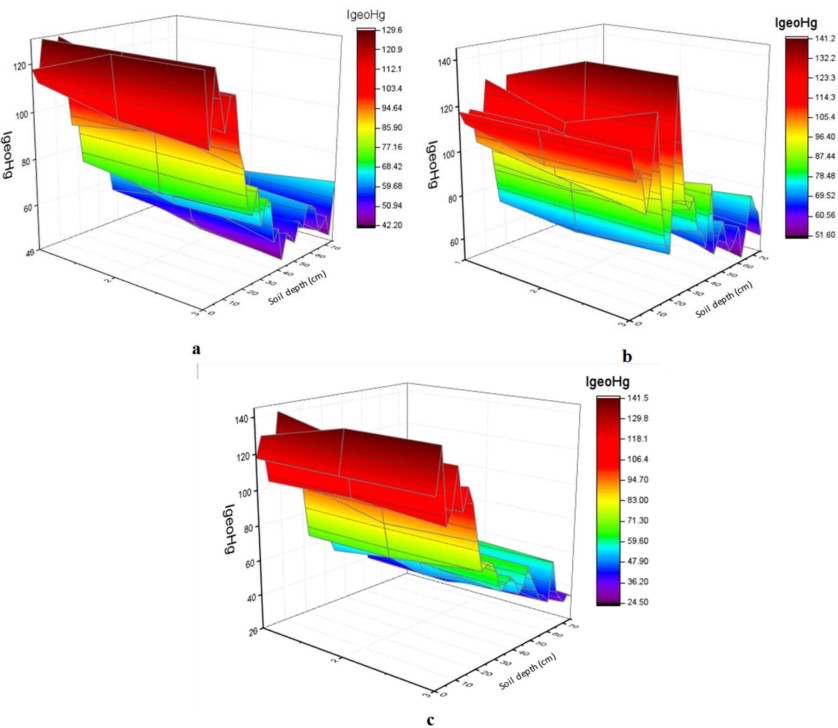

**Figure 5.** Potential risk assessment indices of the soil mercury pollution in grasslands with different degrees of degradation: (**a**). severely degraded; (**b**). moderately degraded; (**c**). non-degraded (1, 2, and 3 in the horizontal axis represent individual sample potential risk assessment index values).

*3.2. Atmospheric Mercury Pollution Analysis and Risk Assessment Results of Different Degraded Grasslands*

3.2.1. Atmospheric Mercury Content Levels in the Different Degraded Grasslands

The monitoring mercury levels in the atmosphere above samples of different degradation types (Figure 6) shows that the atmospheric mercury content levels of the three different degradation levels present a general trend of decreasing in the morning and evening hours, while the mercury content levels increased from noon (12:00) until 2 p.m. (14:00). The mercury content above the severely degraded sample reached the highest value at approximately 14:00, with the highest concentration determined to be 131.90 ng/m$^3$, and it is the highest in comparison to the others. The average value of the mercury content in the moderately degraded sample plots reached the highest value at 12:10, and the highest mercury content was found to be 121.21 ng/m$^3$. On the other hand, the atmospheric mercury content levels over the non-degraded sampling area reached the highest value of 117.09 ng/m$^3$ at approximately 16:00. The average mercury content was the lowest among the three samples in that sampling area, with a trend of higher values in the evening and lower in the early morning. There are no significant differences among the three areas in terms of absolute values. The non-degraded area has a quite distinct temporal trend (maximum at about 17:00).

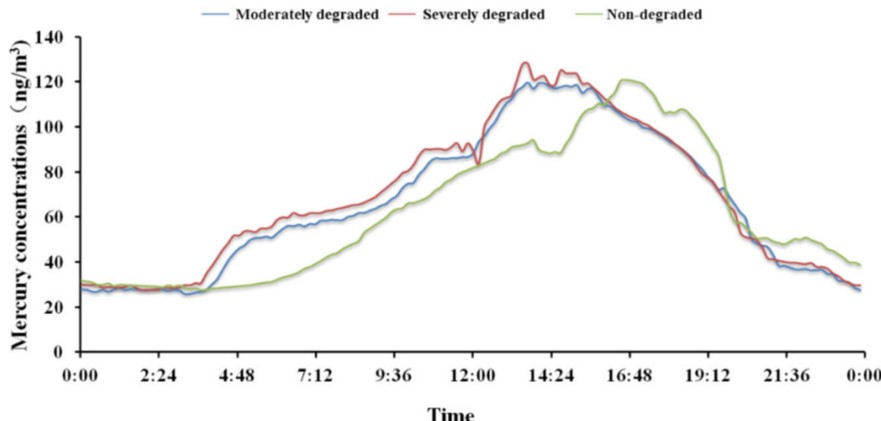

**Figure 6.** Trend chart (24 h) of the atmospheric mercury concentrations in the grasslands, with different degrees of degradation.

3.2.2. Ecological Risk Assessments of the Atmospheric Mercury in the Different Degraded Grasslands

This study used Origin 8.0 to construct the contour maps of atmospheric mercury ecological risk indices of the three grassland types and the results are shown in Figure 7. The potential risk assessment indices of atmospheric mercury of the three grasslands had the same trend of change, which increased first and then decreased, and there was no significant difference among them. Furthermore, it can be seen from Figure 6 that the time range of high risk was mainly concentrated between 11:00 and 16:00.

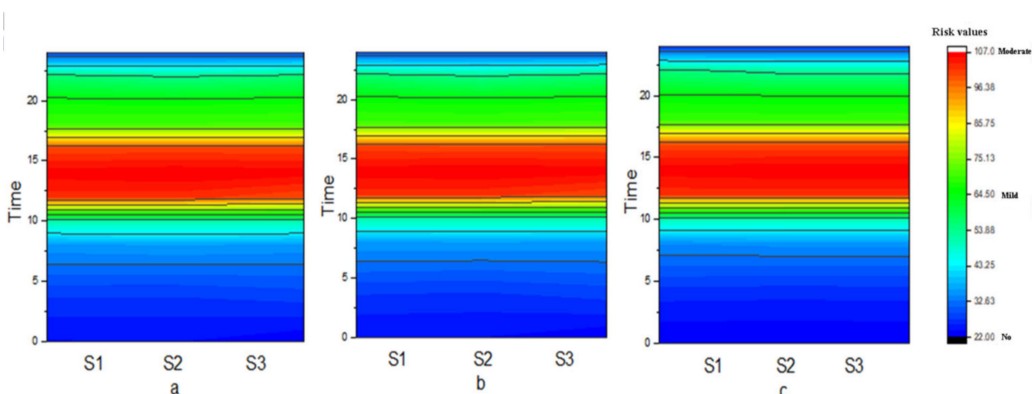

**Figure 7.** Potential risk assessment indices of the mercury levels in the atmosphere with different degrees of degradation: (**a**). severely degraded; (**b**). moderately degraded; (**c**). non-degraded.

*3.3. Linear Correlation Analysis of the Atmospheric and Soil Mercury in the Different Degraded Grasslands*

Linear correlation analysis of the atmospheric and soil mercury in the different degraded grasslands returned the following relationship:

$$\text{Hg}_{\text{soil}} = -0.7857 + 6.987 \times \text{Hg}_{\text{atmospheric}} \ (R = 0.853, p < 0.01) \tag{4}$$

There is a positive correlation between atmospheric mercury and soil mercury. This suggests that part of the soil mercury originates from migration or settlement with the soil atmosphere, which is also related to the particularity of elemental mercury. Soil mercury can derive from adsorption of atmospheric mercury. Conversely, atmospheric mercury may be a re-emission from soil (as strongly suggested by the time trend: more irradiation, more mercury in the atmosphere (Figure 5)).

### 3.4. Analysis and Risk Assessment Results of the Mercury Pollution in the L. chinensis of the Different Degraded Grasslands

3.4.1. Mercury Content Levels in the *L. chinensis* of the Different Degraded Types of Grassland

The mercury concentrations of the roots, below ground stems, upper stems, upper leaves, and lower leaves were measured. The results are shown in Figure 8. The mercury content of herbage was the highest in the severely degraded grassland and the lowest in the non-degraded grassland. In the moderately degraded soil, the mercury content of the *L. chinensis* showed the following pattern: upper leaves > below leaves and upper stems > below stems. Moreover, there were no significant differences observed between the upper leaves and below leaves. In the non-degraded soil, the mercury content levels showed the following pattern: below leaves > upper leaves > roots, and there were no significant differences observed between the upper and below stems. However, in the severely degraded grassland, a pattern of upper leaves > below leaves > roots was observed.

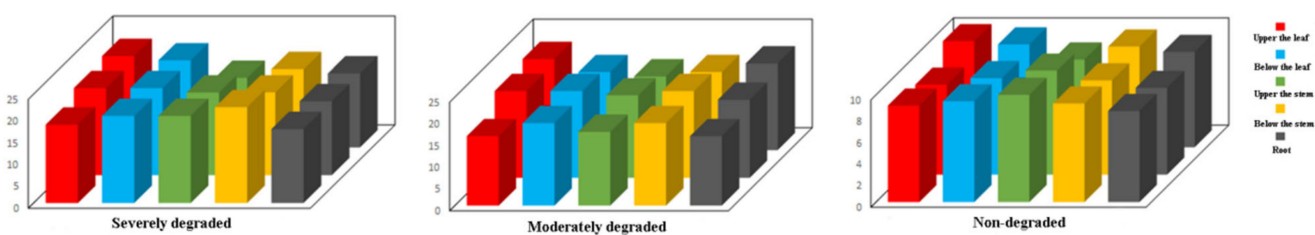

**Figure 8.** Mercury content levels in *L. chinensis* of the different degraded grassland types.

3.4.2. Results of the Neural Network Training

A three-layer evaluation model was applied in order to simulate the spatial distributions and diffusion patterns of the Hg elements, for the purpose of validating the model. Then, the model could be evaluated by comparing the original data with the spatial distribution and diffusion of mercury through interpolation. Compared with the experimental data, it could be seen that the three-layer evaluation model was able to simulate the content levels of the Hg elements in different parts of the *L. chinensis*. It was found that the results were relatively correct and close to the real values, which indicated that the model could meet the requirements of the estimations of mercury concentrations.

The mercury content test set data of the *L. chinensis* in the different degraded grasslands were input into the trained network in order to obtain the prediction results of the model. In the current investigation, based on the results of the simulation and actual monitoring and the content levels of atmospheric mercury and soil mercury in the different degraded grasslands, it was determined that the fitting error range of *L. chinensis* mercury in the moderate degraded grassland was between 3.13% and 18.71%, with an average relative error of 7.1%, which was considered to be relatively good. The fitting error range of the *L. chinensis* mercury was between 2.06% and 21.97%, and the average relative error was 7.93%, which indicated overall good results in the non-degraded grassland. However, the predicted values of most of the mercury sampling points of the *L. chinensis* with severe degradation were found to be significantly different from the measured values, with an average relative error of 20.40% (Figure 9). Therefore, it was considered that the fitting effects were poor.

The generalization ability of mercury content in different parts of *L. chinensis* may be due to the insufficient fitting and prediction of a single BP neural network model, which needs to be optimized. Therefore, neural networks can be successfully applied to the predictions and evaluations of heavy metal content distribution in forage in different types of degraded grasslands, but further exploration will be needed to improve the accuracy. The main advantage of the method proposed in this study was that it avoided the need to model complex processes, such as heavy metal transport and deposition.

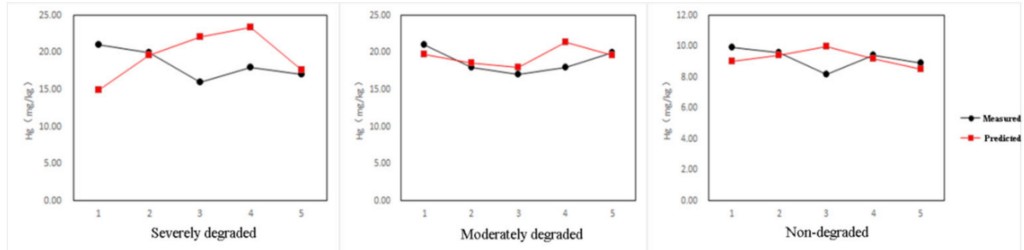

**Figure 9.** Generalization results of the BP neural network. The numbers in the abscissa represent the number of test sample.

## 4. Discussion

*4.1. Mercury Pollution Levels in the Soil of Different Degraded Grassland and the Influencing Factors*

Mercury accumulations in soil may originate from both natural and human factors [40–42]. In recent years, due to overgrazing, over-farming, and over-salinization, grassland areas have been seriously degraded. Consequently, environmental quality and ecological security levels have recently received increasing amounts of attention. This study's comparison of the acquired data with the soil mercury risk assessments of other grasslands (Table 5) revealed that the soil mercury content levels of the Songnen grasslands were higher. The potential risk index assessment levels were higher, and the pollution was more severe.

**Table 5.** Risk assessments of the mercury levels in other grassland areas.

| Area | Hg (mg/Kg) | PI | PI Class | Igeo | Igeo Class | Reference |
|---|---|---|---|---|---|---|
| Typical Grasslands of Tibet | 0.078 | 1.20 | Moderate | −0.32 | Unpolluted | [43] |
| Nagqu Frigid Grasslands | 0.05 | 0.77 | Low | −0.96 | Unpolluted | [44] |
| Nalat Grasslands | 0.01 | 0.15 | Low | −3.29 | Unpolluted | [45] |
| Karajun Grasslands | 0.01 | 0.15 | Low | −3.29 | Unpolluted | [46] |
| Tangbula Grasslands | 0.012 | 0.18 | Low | −3.02 | Unpolluted | [47] |
| Balak Grasslands | 0.013 | 0.20 | Low | −2.91 | Unpolluted | [48] |
| Napa Hyira Grasslands | 0.269 | 4.14 | High | 1.46 | Moderately polluted | [49] |
| Longli Grasslands | 0.686 | 10.55 | High | 2.81 | Moderately to strongly polluted | [50] |
| Xilamuren Grasslands | 0.077 | 1.18 | Moderate | −0.34 | Unpolluted | [51] |
| Xilingol Grasslands | 0.196 | 3.02 | High | 1.01 | Moderately polluted | [52] |
| Erdos Steppe | 0.01 | 0.15 | Low | −3.29 | Unpolluted | [53] |
| Lantern River Grasslands | 0.0284 | 0.44 | Low | −1.78 | Unpolluted | [54] |
| Hohror Grasslands | 0.02 | 0.31 | Low | −2.29 | Unpolluted | [55] |
| Bayan Khusok Grasslands | 0.03 | 0.46 | Low | −1.70 | Unpolluted | [56] |
| Beijing's First Grassland | 0.04 | 0.62 | Low | −1.29 | Unpolluted | [57] |
| Zhang Bei Grasslands | 0.051 | 0.78 | Low | −0.93 | Unpolluted | [58] |
| Qilian Mountain Grasslands | 0.136 | 2.09 | Moderate | 0.48 | Unpolluted to moderately polluted | [59] |
| Gold and Silver Grasslands | 0.418 | 6.43 | High | 2.10 | Moderately to strongly polluted | [60] |
| Yushu Grasslands | 0.046 | 0.71 | Low | −1.08 | Unpolluted | [61] |

Mercury in soil is known to originate from various sources, and the content levels of mercury in soil are mainly affected by those sources. The ecological risks caused by different sources also vary. Mercury is released from a variety of sources through different natural processes [62,63]. Mercury can be transported over some distance in the atmosphere and then returned to the Earth's surface via dry and wet deposition. More than 90% of mercury emissions end up in terrestrial ecosystems, with soil being the largest recipient [62]. Anthropogenic sources, including many industrial point sources, are estimated to release approximately 1960 tons of mercury per year [62,64].

In this study, it was considered that the differences in the risk values of mercury in the grassland soil with different degrees of degradation may also be related to the physical and chemical properties of the soil itself [65], since the degrees of soil degradation led to different content levels of organic matter in the soil. Previous related studies have shown that soil organic matter contains large amounts of functional groups, which form complexes with mercury through complexation and chelation. Organic mercury complexes tend to be very stable [66]. This study found that in the three types of degraded grasslands, both the Hg accumulation index and the potential ecological risk index showed decreasing trends with increasing depth of the sampling points (Figure 4). This may be related to the vertical distribution patterns of the total Hg in the soil, along with changes of organic matter content [67,68]. The reduction in potential risk with depth is mainly due to the reduction in mercury content. This also supports the view that the main source of mercury in the region is long-distance transport of airborne mercury.

### 4.2. Atmospheric Mercury Pollution in the Different Degraded Grasslands and the Influencing Factors

Atmospheric mercury has three forms: gaseous elemental mercury (GEM), particulate bound mercury (PBM), and gaseous mercury oxide (GOM). All of those three forms of mercury can be directly discharged from both anthropogenic and natural sources or converted into each other through chemical transformation and gas particle distribution processes [69,70]. Mercury in the air will eventually be deposited on the Earth's surface via wet and dry deposition and may then be converted to methyl mercury and affect human and ecosystem health [71]. China is currently the largest source of atmospheric mercury in the world [72]. Many studies have been conducted regarding the sources and distribution patterns of mercury in China [8,73]. However, the associations between atmospheric mercury and known anthropogenic sources of mercury emissions have not yet been accurately examined. Northeastern China is an important source of human mercury emissions. The atmospheric mercury is mainly from coal burning and industrial activities [74,75]. A portion of the Songnen Plains' grassland areas is located in the western part of the Jilin Province. The atmospheric mercury sources in that region may be related to the several neighboring cities, such as Changchun, a typical industrial city in Northeast China. The Songnen Plains may be affected by the spring wind direction (southeast wind), resulting in increased atmospheric mercury deposition.

It was observed in this study that the content levels and distribution patterns of the atmospheric mercury over the three different degraded grasslands were not significantly different. This may be related to the migration and transformation rates of atmospheric mercury. The values measured in one day were all found to be below the severe risk level, which may have been partly due to the relatively low regional anthropogenic Hg emission intensity, limited exposure to human sources, and the rapid deposition of the atmospheric Hg to plants [76,77]. Furthermore, the risk values of the atmospheric mercury at different times of the day showed that the mercury content was the highest in early afternoon (14:00), and the ecological risk was also the highest at that time. It was observed that temperature had positive effects on the deposition of the atmospheric mercury and was related to the day and night cycle [31]. Many previous studies have demonstrated that the potential mechanisms driving seasonal and diurnal variations in specific atmospheric mercury concentrations [78,79] may be related to the photosynthesis and physiological ecology of plants.

### 4.3. Mercury Pollution Degrees of the Herbage in the Different Degraded Grasslands and the Influencing Factors

Vegetation provides important land–atmosphere interfaces and is an important factor for the circulation and exchange of atmospheric pollutants and ecosystems. Vegetation also plays a key role in the global atmospheric deposition of mercury pollution with a long atmospheric residence time [80,81]. In individual sites, especially in forests, long-term studies have reported that the uptake of atmospheric mercury by ground plant tissues

contributes significantly to atmospheric deposition [82]. The soil mercury content levels were closely related to vegetation types, with the lowest mercury concentrations in barren areas (such as deserts) and the highest mercury concentrations in densely vegetated areas [83,84]. A recent study found a significant correlation between Hg concentrations and plant photosynthetic activity indices at many northern hemisphere monitoring sites, with plant photosynthetic rates declining in winter and rising in summer. Those findings suggest that the enhanced dry deposition of Hg on vegetation canopies may lead to decreased Hg concentrations during warmer seasons [81]. Since the dry deposition rates of heavy metals on vegetation surfaces are also higher than that on non-vegetation surfaces, the dry deposition of heavy metals on vegetation canopies caused by seasonal vegetation activities may also be an important reason for the seasonal changes in heavy metal concentrations [80].

The current annual re-emission of mercury is estimated at 4000 to 6300 tons/year [62]. A large part of the re-emission of mercury may eventually accumulate in the topsoil. In the non-degraded grassland area examined in this study, it was observed that a large amount of dense growth of forage grass concentrated large amounts of mercury. This may also explain why the non-degraded grasses show lower mercury content in soil than the other two types of grassland. In addition, studies have shown that the mercury sources found in the aboveground tissues of vegetation are mainly from atmospheric mercury absorption [85]. However, the mechanism of mercury absorption by plants is still unclear, but it can be derived from atmospheric mercury deposition in plant leaves. Therefore, the mercury content levels of forage grass in different degraded grasslands may also show variations.

*4.4. Transport and Transformation of Mercury in the Atmosphere-Soil-Herbage of the Songnen Plains Grasslands and the Influencing Factors*

In the current study, the migration of mercury in the examined three types of grassland was mainly from the atmosphere to the soil, and then from the soil to plants. Since no plant species have been identified as super accumulators of mercury at the present time, it is generally believed that high biomass is the key factor for mercury enrichment in plants [5]. Therefore, it can be inferred that the elemental mercury content in the soil of non-degraded grasslands may be lower due to higher biomass coverage. Moreover, due to the larger biomass and high vegetation coverage, the average amount of mercury distributed to each plant was also lower. In addition, severely degraded meadows near bare land had few surface plants. However, mercury contents in plants in the non-degraded grassland look quite similar to the severely degraded grassland; this shows that mercury in soil does not show excessive enrichment in plants, which may also be related to the species of plants and their ability to enrich mercury in soil.

The mercury content levels of the *L. chinensis* in the different types of grassland show that with the deepening of the degradation, the mercury content in roots also increased. Those findings indicated that, within a certain range, the mercury content in the soil was positively correlated with the mercury content in the plants. Since Hg is not a plant micronutrient, not many plants with a high concentration of mercury have been identified at this time [86]. However, it is considered that the dominant species *L. chinensis* is the main plant that enriches mercury in the grasslands of the Songnen Plains region.

The high volatility of mercury may lead to air pollution, which in turn leads to secondary pollution. At the same time, biological exposure to mercury may occur during the process of plant consumption. For example, the grasses from the Songnen Plains may be eaten by cattle, sheep, insects, etc., and mercury pollutants may enter the food chain through herbivores [87,88]. At the present time, few studies have reported that mercury will be volatilized into the atmosphere again after entering plants [65]. Therefore, mercury transfer between soil and plants is an important part of the mercury cycle in the Songnen grasslands.

## 5. Conclusions

The results obtained in this study shed some light on the distribution patterns of mercury content in grasslands characterized by different degrees of degradation. In this study, the moderately degraded grassland soil displayed the largest differences. Meanwhile, the mercury content in the non-degraded grassland soil was observed to be smaller than that in the other grasslands. The risk assessment results of soil mercury revealed that the potential risk indices of soil mercury were the highest in the severely degraded and moderately degraded grassland areas. The lowest potential risk appears to be associated with severely degraded soil, and the other two types do not show appreciable differences. It was found that the potential risks in the non-degraded grassland of atmospheric mercury were low, and the potential values were reduced with increases in time. In regard to the moderately degraded grassland, the potential risks of the atmospheric mercury were below a moderate risk level. Moreover, in the cases of slight pollution during daytime, the risk values showed a decreasing trend after the first increase. The occurrence times of the highest risk values were different for different types of grassland. High atmospheric mercury content was associated with high mercury content in the soil, and there was a significant positive correlation between them. Those findings suggested that some of the mercury in the soil may be transported or deposited through the atmosphere. The results of Hg neural network simulations for the *L. chinensis* in the different types of grassland indicated that there were only minimal differences between the predicted values and the measured values, with a certain degree of accuracy.

**Author Contributions:** Conceptualization, Z.W. and G.Z.; methodology, X.L. (Xu Li), Y.Z., and X.Z.; software, Y.W. and L.W.; validation, G.Z. and L.W.; formal analysis, Z.T., D.W., and Z.F.; data curation, X.L. (Xiangyun Li), M.C., and T.X.; writing—original draft preparation, Z.W. and L.W.; writing—review and editing, G.Z. and L.W.; visualization, L.W.; supervision and project administration, G.Z.; funding acquisition, Z.W. All authors have read and agreed to the published version of the manuscript.

**Funding:** This research was funded by the National Natural Science Foundation of China (31230012, 31770520) and the Key Social Development Project of the Jilin Science and Technology Department of China (20190303068SF).

**Institutional Review Board Statement:** Not applicable.

**Data Availability Statement:** The data presented in this study are available on request from the corresponding author.

**Acknowledgments:** We are grateful to the Key Laboratory of Vegetation Ecology of the Ministry of Education for its help and support.

**Conflicts of Interest:** The authors declare that they have no known competing financial interests or personal relationships that could have appeared to influence the work reported in this paper.

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
