# Peer review of "Characteristics of Mercury Pollution and Ecological Risk Assessment in Different Degraded Grasslands of the Songnen Plains, Northeastern China"

_sustainability, doi:10.3390/su131910898_

Round 1
Reviewer 1 Report
Review of the paper Characteristics of Mercury Pollution and Ecological Risk Assessments in Different Degraded Grasslands of the Songnen Plains
The paper describes mercury contamination in grasslands of Northeastern China. The topic is potentially interesting, but the current presentation quality is low.
The English form is inadequate and requires editing by a native speaker. I made a number of suggestions in the attached file, but there is much more!
The text is lengthy, very confusing, and lacks of focus; the introduction, discussion, and conclusions are engulfed with many statements that are just repetitions from literature, and blur anything new may be in the paper. Try being clearer and more concise.
Which is the motivation of the study (i.e., why in that specific area, and not in other grasslands?)? Was mercury contamination in the area previously known? (references?) Which is (are) the source(s)?
There is a serious lack of information on analytical methods (standards? accuracy? precision?).
Fig. 1 should be complemented by a smaller scale map of the specific study area, showing the location of sampling points. Fig. 3 is essentially useless, whereas an additional Figure showing boxplots of concentration values could be useful. Figs. 5 and 8 should be improved (see comments in the attached file)
It is ESSENTIAL that the full analytical dataset be provided as a supplementary file.
Discussion and conclusions are mostly descriptive, plagued (as stated above) by many unnecessary statements from the literature, and do not convey any new insight. A key point is that there is no attempt to explain what could be the actual mercury source(s), and what relationship exists (possibly none?) between mercury contents and degradation status. The level of pollution is indeed moderate, so it’s no surprise that there are no substantial risk differences. I cannot understand the usefulness of the neural network section (admittedly, I am not familiar with these techniques, but it seems to me that in this case, the consequences are quite light).
More details are in the attached pdf.

Author Response
From: Gang Zhang
School of Environment, Northeast Normal University,
Key Laboratory of Vegetation Ecology, Ministry of Education, Northeast Normal University,
Institute of Grassland Science, Northeast Normal University,,
Changchun Jilin, 130117, China.
Editor in Sustainability
September 7 2021
Thank you for considering the revised version of our manuscript “Characteristics of Mercury Pollution and Ecological Risk Assessment in Different Degraded Grasslands of the Songnen Plains, Northeastern China (Ms. ID.: sustainability-1339792)”, for publication in Sustainability. We appreciate all the comments from the reviewers on the previous manuscript. We have thoughtfully considered these comments. The following are the itemized responses to the reviewers’ comments (given in blue below). We highlighted all the changes by using a red font in the revised manuscript.
In addition to the reviewers’ revision suggestions, we also found some other shortcomings in the manuscript and have corrected them in the revised manuscript. We also highlighted them by using a red font.
Thank you very much!
Yours sincerely,
Gang Zhang behalf of the authors.
Major points: (“C” means Comment, “R” means Response)
C1: The topic is potentially interesting, but the current presentation quality is low.
R1: Now, We have carefully revised and polished the article.
C2: The English form is inadequate and requires editing by a native speaker. I made a number of suggestions in the attached file, but there is much more!
R2: This article has been polished by native English speakers, and the polishing certificate is attached at the end of the article. As for some suggestions put forward by reviewers, we have modified the manuscript, and the modified part has been marked in red font.
C3: The text is lengthy, very confusing, and lacks of focus; the introduction, discussion, and conclusions are engulfed with many statements that are just repetitions from literature, and blur anything new may be in the paper. Try being clearer and more concise.
R3: Thank you for your advice. For this problem, we have revised the content of the manuscript to make the whole article clearer.
C4: Which is the motivation of the study (i.e., why in that specific area, and not in other grasslands?)? Was mercury contamination in the area previously known? (references?) Which is (are) the source(s)?
R4: Our main motivations for choosing this area as the study area include:
- There is little human activity near the study area, and there is no large-scale mining and industrial production area;
- The grassland in the study area is typical temperate grassland in Eurasia;
- This area represents the current situation of most grassland in China, mainly grazing, and represents the research on mercury in grassland background area.
Our aim is to address mercury cycling processes in large areas of grassland that are unpolluted and heavily disturbed and utilized by humans in the context of the global mercury cycle, with typical demonstration significance. Another important reason is that we have set up a grassland positioning experiment station in the research area, which facilitates the selection of the research area. At present, there is no relevant research report in China to carry out continuous location observation of single mercury element in grassland, and we have carried out a series of studies for more than 5 years. The main sources of mercury in this grassland are atmospheric mercury deposition and background mercury in soil due to low disturbance degree.
C5: There is a serious lack of information on analytical methods (standards? accuracy? precision?).
R5: In this study, all samples were analyzed by the Lumex RA915+ analyzer using its solid and liquid attachment. This machine measures gaseous mercury by: High frequency modulation polarized zeeman effect background correction techniques, to 254 nm resonance emission line of ground state mercury atom absorption is analyzed. Determination of solid mercury (e.g. soil mercury) by thermal analysis (in accordance with EPA7473 method). We have developed mature experience in the application of this analysis method and have published relevant articles, such as Wang et al., 2020. (Pollution Characteristics and Risk Assessments of Mercury in the Soil of the Main Urban Regions in a Typical Chinese Industrial City: Changchun); Feng et al., 2020. (Mercury spatial distribution characteristics and its exposure of the endangered Jankowski's bunting). We think this analytical method can be used.
C6: Fig. 1 should be complemented by a smaller scale map of the specific study area, showing the location of sampling points. Fig. 3 is essentially useless, whereas an additional Figure showing boxplots of concentration values could be useful. Figs. 5 and 8 should be improved (see comments in the attached file)
R6: The location of our research site has been marked in Figure 1 (red pentacle). We have removed Figure 3. We have modified Figure 5 and Figure 8 according to suggestions of reviewers. Please refer to the figure in the text.
C7: It is ESSENTIAL that the full analytical dataset be provided as a supplementary file.
R7: We have supplemented the neural network analysis steps and procedures as attachments.
C8: Discussion and conclusions are mostly descriptive, plagued (as stated above) by many unnecessary statements from the literature, and do not convey any new insight. A key point is that there is no attempt to explain what could be the actual mercury source(s), and what relationship exists (possibly none?) between mercury contents and degradation status. The level of pollution is indeed moderate, so it’s no surprise that there are no substantial risk differences. I cannot understand the usefulness of the neural network section (admittedly, I am not familiar with these techniques, but it seems to me that in this case, the consequences are quite light).
R8: Because the research around it and there is no obvious source of mercury, so we when conducting the research mainly consider the global mercury cycle. Since the industrial Revolution, the amount of mercury in atmospheric mercury reservoirs has increased, mercury cycling around the world may be deposited on a global scale, and this effect increases the risk of mercury exposure in soil and vegetation, thus increasing the amount of mercury in soil and vegetation. Although these levels are moderate, the long-term ecological risks of mercury are of concern. We use neural network mainly is to build a model to predict mercury content in plants and its mainly through the external environment factors (such as we measured the soil mercury levels, atmospheric mercury levels, etc.) to predict the amount of mercury in plants and to establish a nonlinear mapping relationship, so can greatly reduce the samples directly to the plant for testing, and MATLAB or other software can be used to predict directly.
Attachment: English polishing certificate

Reviewer 2 Report
- Extensive abstract, tiresome with little objectivity, it is suggested that authors be more concise.
- the text content of the topic Conclusion is very similar to the Abstract. Authors should note what they really emphasize as the original contribution obtained in the work.
- It is suggested that the standard used for the procedure mentioned in lines 271 to 276 be cited.
- Clarify how the data analysis method is validated to ensure the reliability of the simulation results with the collected data.
Author Response
From: Gang Zhang
School of Environment, Northeast Normal University,
Key Laboratory of Vegetation Ecology, Ministry of Education, Northeast Normal University,
Institute of Grassland Science, Northeast Normal University,,
Changchun Jilin, 130117, China.
Editor in Sustainability
September 7 2021
Thank you for considering the revised version of our manuscript “Characteristics of Mercury Pollution and Ecological Risk Assessment in Different Degraded Grasslands of the Songnen Plains, Northeastern China (Ms. ID.: sustainability-1339792)”, for publication in Sustainability. We appreciate all the comments from the reviewers on the previous manuscript. We have thoughtfully considered these comments. The following are the itemized responses to the reviewers’ comments (given in blue below). We highlighted all the changes by using a red font in the revised manuscript.
In addition to the reviewers’ revision suggestions, we also found some other shortcomings in the manuscript and have corrected them in the revised manuscript. We also highlighted them by using a red font.
Thank you very much!
Yours sincerely,
Gang Zhang behalf of the authors.
Major points: (“C” means Comment, “R” means Response)
C1: Extensive abstract, tiresome with little objectivity, it is suggested that authors be more concise.
R1: We have revised the whole article to make it more concise and clear.
C2: the text content of the topic Conclusion is very similar to the Abstract. Authors should note what they really emphasize as the original contribution obtained in the work.
R2: Thank you for your advice. We have rewritten the abstract and conclusion.
C3: It is suggested that the standard used for the procedure mentioned in lines 271 to 276 be cited.
R3: Thank you for your advice. We have cited in this section the article on the use and introduction of the machine. “Feng Z, Xiao T, Zhang G, et al. Mercury spatial distribution characteristics and its exposure of the endangered Jankowski's bunting [J]. Ecological Research, 2020, 35:650-661.”
C4: Clarify how the data analysis method is validated to ensure the reliability of the simulation results with the collected data.
R4: We use neural network mainly to establish a model to predict mercury content in plants, which mainly predicts mercury content in plants through external environmental factors (such as soil mercury content and atmospheric mercury content measured by us), so as to establish a nonlinear mapping relationship and finally get the predicted value of mercury content in plants. At the same time, we directly collected plant samples in the plot and brought them back to the laboratory for mercury analysis. The measured values and predicted values in our experiment were compared, which were explained in Section 3.4.2. The predicted values and experimental values are shown in Figure 8. At present, methods and technologies of using BP artificial neural network to analyze and predict heavy metals have been widely used, such as: Derin, et al. 2016. (Phosphate, Phoshide, Nitride and Carbide Capacity Predictions of Molten Melts by Using an Artificial Neur Network Approach. ISIJ International, 2016(2): 183-188.); Li, et al. 2011. (Study on spatial distribution of soil heavy metals in Huizhou city based on BP-ANN modeling and GIS. Procedia Environmental Sciences, 1953-1960.); Jia, et al. 2018. (Comparison Study on the Estimation of the Spatial Distribution of Regional Soil Metal(loid)s Pollution Based on Kriging Interpolation and BP Neural Network. International Journal of Environmental Research and Public Health, 34-48).

Reviewer 3 Report
In this manuscript the migration and transformation of Hg from the atmosphere to the soil and the soil to the plants was studied. The results were studied using statistical methods and several hypothesis were considered.
The paper may be accepted because is relevant and provides a solid framework of researches that can be employed in the future.
Author Response
From: Gang Zhang
School of Environment, Northeast Normal University,
Key Laboratory of Vegetation Ecology, Ministry of Education, Northeast Normal University,
Institute of Grassland Science, Northeast Normal University,,
Changchun Jilin, 130117, China.
Editor in Sustainability
September 7 2021
Thank you for considering the revised version of our manuscript “Characteristics of Mercury Pollution and Ecological Risk Assessment in Different Degraded Grasslands of the Songnen Plains, Northeastern China (Ms. ID.: sustainability-1339792)”, for publication in Sustainability. We appreciate all the comments from the reviewers on the previous manuscript. We have thoughtfully considered these comments. The following are the itemized responses to the reviewers’ comments (given in blue below). We highlighted all the changes by using a red font in the revised manuscript.
In addition to the reviewers’ revision suggestions, we also found some other shortcomings in the manuscript and have corrected them in the revised manuscript. We also highlighted them by using a red font.
Thank you very much!
Yours sincerely,
Gang Zhang behalf of the authors.
Major points: (“C” means Comment, “R” means Response)
C1: In this manuscript the migration and transformation of Hg from the atmosphere to the soil and the soil to the plants was studied. The results were studied using statistical methods and several hypothesis were considered.
C2: The paper may be accepted because is relevant and provides a solid framework of researches that can be employed in the future.
R: Thank you for your review. We found some other shortcomings in the manuscript and have corrected them in the revised manuscript. We highlighted them by using a red font.

Reviewer 4 Report
I only have a few comments on the work:
There is no source for line 93.
Figure 1: please specify the location (a dot would be enough).
Line 353: Why moderately degraded soil contain more mercury than severely degraded? Can you clarify that?
Author Response
From: Gang Zhang
School of Environment, Northeast Normal University,
Key Laboratory of Vegetation Ecology, Ministry of Education, Northeast Normal University,
Institute of Grassland Science, Northeast Normal University,,
Changchun Jilin, 130117, China.
Editor in Sustainability
September 7 2021
Thank you for considering the revised version of our manuscript “Characteristics of Mercury Pollution and Ecological Risk Assessment in Different Degraded Grasslands of the Songnen Plains, Northeastern China (Ms. ID.: sustainability-1339792)”, for publication in Sustainability. We appreciate all the comments from the reviewers on the previous manuscript. We have thoughtfully considered these comments. The following are the itemized responses to the reviewers’ comments (given in blue below). We highlighted all the changes by using a red font in the revised manuscript.
In addition to the reviewers’ revision suggestions, we also found some other shortcomings in the manuscript and have corrected them in the revised manuscript. We also highlighted them by using a red font.
Thank you very much!
Yours sincerely,
Gang Zhang behalf of the authors.
Major points: (“C” means Comment, “R” means Response)
C1: There is no source for line 93.
R1: Thank you for your advice. We have cited relevant article in this section.
C2: Figure 1: please specify the location (a dot would be enough).
R2: Thank you for your advice. The location of our research site has been marked in Figure 1 (red pentacle).
C3: Line 353: Why moderately degraded soil contain more mercury than severely degraded? Can you clarify that?
R3: At present, there are few studies on soil mercury in different degraded grasslands, and there are no similar results. But we observed after a long time that the fact is really such, (we have grassland station set up in the area), is indeed a moderate degraded grassland degeneration of soil mercury than the soil mercury content is high, because there is no other reports of the results, so we speculated that the main reason of this result is: The soil mercury in moderately degraded grassland mainly comes from atmospheric mercury deposition, and due to the lack of above-ground vegetation (compared with that in non-degraded grassland), the soil mercury is less absorbed by plants, and most of the soil mercury is retained in the soil, while most of the soil mercury in non-degraded grassland is absorbed by vegetation in plants. And severe degradation by comparison, moderately degraded grassland with plant litter cover (litter after more than 10 years), the surface of the exposed part is less, the mercury in the atmosphere and soil between the circulation is limited by the vegetation, plants absorb mercury as part of the soil retaining in the plant litter, and cannot be absorbed by plants. However, severely degraded soil mercury is easily returned to the atmosphere under sunlight, which accelerates the circulation of severely degraded grassland mercury between soil and atmosphere. This is an interesting phenomenon that deserves further study.

Round 2
Reviewer 1 Report
The revised version addresses only in part the issues raised in the previous round of review. Many annotations on the pdf file were apparently overlooked; this may be due to the difficulty of going through the many notes in the pdf file. For this reason, I will try to list explicitly in the following as many comments as possible; I also tried to express my remarks in more detail, with the hope that this helps to understand what I am trying to say. The paper still has to go a long way before being acceptable for a high-rank international journal (a further round of review may be necessary). Please take your time to carefully consider the following.
General comments
- English language. Despite the alleged check by native speakers, some aspects are still puzzling. For instance: line 15, “health human”; I would say “human health”, right? Lines 22-34: all sentences are separated by semicolons, which seems excessive to me, especially when the sentences deal with distinct, separate concepts. In any case, you should NOT use capital letters after a semicolon. In general, I find that you use too many words: try being more concise. For instance (line 18): “grasslands under different degradation degree type conditions”: unnecessarily redundant: why not simply “different types of grassland” (reference to degradation is just in the previous sentence). A more concise style makes communication more effective. PLEASE CHECK FURTHER SUGGESTIONS IN THE PDF FILE
- There are responses to my remarks that apparently produced no change in the text. For instance, the Introduction remains lengthy and wandering, with many unnecessary sentences; motivations for the study (R4) are not clearly exposed in the text.
- It is mandatory to provide objective data on analytical quality. I agree that the Lumex instrument may be adequate for such a study (we have one in our lab), and you say it was calibrated (how?). However, you must give credible estimates of accuracy and precision by reporting blanks, analysis of certified international standards, and replicate analyses. Specify also detection limits (reported only for plants).
- The supplementary file contains the script for the neural network model – it contains several Chinese characters that will be of no use for most readers. This file does not comply with my request to provide the FULL DATA SET (e.g., as an Excel file). It is a basic requirement of open science to provide original data so that they can be independently evaluated. Indeed, the journal website recommends that Full experimental and methodical details must be provided so that the results can be reproduced. Moreover, in this case several figures do not show enough detail to appreciate differences (see specific comments below)
- There are many confusing, and even contradictory, statements; see specific comments below
- Overall, the paper lacks focus: its message is not clear. If I understand correctly, the goal of the paper is to investigate the (quasi)natural biogeochemical cycle of mercury in an area with little anthropic influence (except for the mercury source, which is supposedly long-range transport of presumably anthropogenic mercury). So where is the connection with sustainability (the focus of the journal)? In my perception, sustainability is the compatibility between human activities and a healthy environment. Apparently, the different degrees of grassland degradation are NOT connected with the presence of mercury, but are a consequence of other cause(s) – principally, salinization, if I understand correctly. So why putting throughout the paper all this emphasis on the different types of grassland? Mercury contents are fairly similar, which is consistent with a long-distance mercury source. The only remarkable difference is the daytime trend of atmospheric mercury, which deserves more discussion (my guess is that in the non-degraded grassland the vegetation cover delays the impact of solar radiation on soil – but it is just a guess). There is another point that definitely requires more discussion. At this stage, we don’t know which is the Hg concentration in your soil, because you report concentrations in ug/kg, which means nothing (see comment to § 3.1.1 below). If it is mg/kg, the content is quite high for a “normal” soil – in many countries, these soils would be considered seriously polluted; indeed you say that such a content exceeds Chinese standards and is much higher than other grasslands. So there must be an important mercury source (or more). You vaguely refer to “surrounding cities” – however no city is shown in Fig. 1, and more important there is no specific information on their role as mercury sources (size? Presence of industries?)
Specific comments
SEE LINGUISTIC REMARKS IN THE PDF
Lines 25-27: as you write it here, there is a contradiction between single factor and accumulation indexes (which suggest a slightly polluted environment), and the potential risk index, pointing to a severe pollution. You should comment on reasons for such a discrepancy, and make clear your stand (do you believe that the area is slightly polluted, or severely polluted?). See further remarks on the confusing, in places contradictory, use of "light", "moderate" "severe"....
Line 64: plants roots adsorb [mercury] through transpiration? Is it so?
Line 135: “geographical location”: I suppose you mean the local geomorphology
Line 139: “due to low disturbance degree”: unclear meaning – how can a low disturbance degree be responsible for the presence of mercury in soil?
Line 155: GB/T 37067-2018: what does this acronym mean? Please describe in full the criteria for assigning the degradation status
Line 163: “heavy soil capacities”: cannot understand what does this mean
Lines 166-167: why was moderately degraded grassland given priority?
Line 204: see general comment #3
Line 207: “after treatment”: what kind of treatment?
3.1.1 and following – you report concentrations in ug/kg, which means nothing – is it mg (micrograms)? mg (milligrams)?
Lines 276-278: indeed values for the non-degraded grassland have a much larger SD – any explanation for this? I suggest you provide also median values. As suggested in the previous review, a figure showing boxplots would be helpful
Lines 279-280: given the SD, the three averages are not really different
Table 2: is n = 48 for all three grassland types?
3.1.2: please report the formulas for calculation of the indexes, and which is the difference between the various indexes. Please explicit the criteria for defining pollution “light”, “mild” etc
Lines 292-298: most indexes correspond to “mild pollution”, however for the cumulative index of the severely degraded soil you speak of “light pollution” – which is the difference?
Lines 309-315: reasoning here is pretty confusing. You start saying that “the moderate degradation grassland had the highest potential risk index”; from what I can see in Fig. 4, the maximum value of Igeo is 141.2 for the moderately degraded soil, and 141.5 for the severely degraded soil. Then you say “the potential risk values of the soil with moderate degradation were lower than the moderate risk values” – which are these “moderate risk values”?
Lines 310 and 315: I suppose that the decrease of the potential risk with depth corresponds to a decrease in the mercury content. This would support the idea that the main source of mercury in the area is long-distance transport of airborne mercury. You may want to comment on this.
Fig. 4: one of the horizontal axes has no label (that with 2-3 marks); what does it represent?
Lines 328-334: I do not feel the differences in maximum values are really significant – which is the experimental uncertainty (see general comment #3….)? From what you show in Fig. 5, the maximum value over the severely degraded soil corresponds to a spike at about 14:00 (not noon as written in the text). The general trend is perfectly overlapping with the moderately degraded area, and the maximum spike could be just a transient event (such as a wind gust – typical in this kind of measurements…). What you can safely say is a) there are no significant differences among the three areas in term of absolute values; b) the non-degraded area has a quite distinct temporal trend (maximum at about 17:00 – not 16:00, at least from what I see in Fig. 5). Can you suggest a reason for this difference?
3.2.2: all this section is quite confusing. You speak about 3D maps – where is the third dimension in Fig. 6? First you say (line 347-348) that the conditions are “under severe risk”, then you speak (lines 355-356) of “moderate risk” and “slightly polluted level” (once and again please explain clearly the criteria for defining “slight”, “moderate”, “severe”….). From Fig. 6, the patterns for the three grassland types look pretty much the same: how can you say that the moderately degraded grassland has the highest potential risk (line 350), and that “the time range of the highest risk value changed with the different grassland types” (line 360-361)? Lines 351 and following: How can the risk be reduced by an increase of the measuring time?
3.3, eqn (4): please give the statistical significance (p value)
Lines 370-371: the correlation works in both ways: soil mercury can derive from adsorption of atmospheric mercury; conversely, atmospheric mercury may be re-emission from soil (as strongly suggested by the time trend: more irradiation, more mercury in the atmosphere)
Line 377: graphs of Fig. 7 does not make easy to see differences (the figure should be complemented by the full set of numerical values, as repeatedly recommended). I can only see that in the non-degraded grassland the overall content of mercury in plants is lower than in the other two types, but other differences are not apparent
Lines 403-405: here do you refer to non-degraded grassland?
Lines 417-418: you say “the sources of the pollution could be located” – where and what? There is nothing in this respect
Lines 426-430: this information is more adequate for the Introduction
4.1: this section has a weak logical structure. It starts with a comparison with other grassland areas (lines 430-433); it is a pure list with no discussion (reason(s) for the differences?). Then there are very general statements (lines 435-442) about mercury sources/transport: what is the connection with this paper? finally (lines 443-453) some speculations about the role of organic matter (you provide NO data on the organic content of soils). There is no obvious connection between the three parts.
Line 433: here you define the pollution “severe”, elsewhere you say “moderate”, “slight”…. Please be clear and consistent!
Line 470: there are no cities shown in Fig. 1. Please indicate the largest centers. Are there data on mercury emissions from these cities? Are they likely sources in terms of wind directions?
Lines 471-479: quite general statements with little demonstrable connection to this paper
Line 488: from Fig. 5 it seems that the highest mercury content occurs at about 14:00
Lines 489-490: unclear what you mean with “positive effects of temperature on deposition of mercury”; do you mean there is a positive correlation?
Lines 491-493: what has this to do with this paper? No data presented in this respect.
Lines 501-502: “This deposition occurs….”: unclear sentence. In principle, atmospheric deposition occurs on living plants and bare soil (it may also occur on a piece of dead wood, of course, but in general it is negligible). When leaves fall, or the entire organism dies, mercury is transferred to the soil. Overall, this sentence adds nothing, and can be removed
Lines 504-515: once and again, a sequence of general statements without specific connection to data presented in this paper
Lines 520-521: the average content in the non-degraded grassland is lower than in the other two types. Moreover, as pointed out before, differences are small and probably not significant.
Line 525: “as a result” of what? How can stomatal absorption result in variations? See also comment to Line 377.
Lines 533-536: oversimplification of rather complex processes for which you have little control. First, as previously said, mercury differences in the three types of soil can hardly be considered significant; second, the conclusion that “more vegetation coverage = less mercury in individual plants” implies that the limiting factor is the atmospheric concentration of mercury: maybe so, but it is just a hypothesis
Line 538: from Fig. 7, Hg contents in plants in the moderately degraded grassland look quite similar to the severely degraded grassland
Lines 544-545: “suitable species” for what?
Lines 545-546: L. chinensis is the only species you studied. You can say nothing about others.
Lines 553-554: with “transformation” you mean entering the food chain?
Line 558: from Table 2, the largest difference is for non-degraded grassland.
5 (Conclusions): reiterates many questionable statements addressed in the previous comments. This section should be rewritten after addressing the above points.
Lines 558-559: “vegetation degradation may affect mercury cycling”: there is nothing in this paper that supports such a conclusion
Lines 559-560: from Table 2, the lowest average Hg content is for non-degraded soil
Lines 561-563: from Fig. 4, the lowest risk appears associated with severely degraded soil; the other two types do not show appreciable differences
Lines 564 and following: light pollution, low risk, moderate risk… please define clearly!
Author Response
From: Gang Zhang
School of Environment, Northeast Normal University,
Key Laboratory of Vegetation Ecology, Ministry of Education, Northeast Normal University,
Institute of Grassland Science, Northeast Normal University,,
Changchun Jilin, 130117, China.
September 17 2021
Dear Editor and Reviewer in Sustainability
Thank you for considering the revised version of our manuscript “Characteristics of Mercury Pollution and Ecological Risk Assessment in Different Degraded Grasslands of the Songnen Plains, Northeastern China (Ms. ID.: sustainability-1339792)”, for publication in Sustainability. We appreciate all the comments from the reviewers on the previous manuscript. We have thoughtfully considered these comments. The following are the itemized responses to the reviewers’ comments (given in blue below). We highlighted all the changes by using a red font in the revised manuscript.
In addition to the reviewers’ revision suggestions, we also found some other shortcomings in the manuscript and have corrected them in the revised manuscript. We also highlighted them by using red font (first round) and blue font (second round).
Thank you very much!
Yours sincerely,
Gang Zhang behalf of the authors.
Major points: (“C” means Comment, “R” means Response)
Reviewer comments
General comments
C1: English language. Despite the alleged check by native speakers, some aspects are still puzzling. For instance: line 15, “health human”; I would say “human health”, right?
R1: Yes, We have changed it. “Health human” changed to “human health”.
C2: Lines 22-34: all sentences are separated by semicolons, which seems excessive to me, especially when the sentences deal with distinct, separate concepts. In any case, you should NOT use capital letters after a semicolon. In general, I find that you use too many words: try being more concise. For instance (line 18): “grasslands under different degradation degree type conditions”: unnecessarily redundant: why not simply “different types of grassland” (reference to degradation is just in the previous sentence).
R2: Lines 22-34 have been modified and the semicolon has been changed to a period. We've tried to make the article more concise. Line 18: “grasslands under different degradation degree type conditions” to “different types of grassland”.
C3: A more concise style makes communication more effective. PLEASE CHECK FURTHER SUGGESTIONS IN THE PDF FILE
R3: Thanks for your advice. We have made detailed modifications according to the suggestions in the PDF file.
C4: There are responses to my remarks that apparently produced no change in the text. For instance, the Introduction remains lengthy and wandering, with many unnecessary sentences; motivations for the study (R4) are not clearly exposed in the text.
R4: We have already described our motivation in the last paragraph of the introduction.
C5: It is mandatory to provide objective data on analytical quality. I agree that the Lumex instrument may be adequate for such a study (we have one in our lab), and you say it was calibrated (how?). However, you must give credible estimates of accuracy and precision by reporting blanks, analysis of certified international standards, and replicate analyses. Specify also detection limits (reported only for plants).
R5: Thanks for your advice. We have added a new section in 2.5 to introduce quality control for instrument proofreading.
C6: The supplementary file contains the script for the neural network model – it contains several Chinese characters that will be of no use for most readers. This file does not comply with my request to provide the FULL DATA SET (e.g., as an Excel file). It is a basic requirement of open science to provide original data so that they can be independently evaluated. Indeed, the journal website recommends that Full experimental and methodical details must be provided so that the results can be reproduced. Moreover, in this case several figures do not show enough detail to appreciate differences (see specific comments below)
R6: Thanks for your advice. We have modified the Chinese characters in the attachment and we cannot provide the original data. The calculation can be completed according to the procedures in the annex. We don't think it is necessary to provide the original data.
C7: Overall, the paper lacks focus: its message is not clear. If I understand correctly, the goal of the paper is to investigate the (quasi)natural biogeochemical cycle of mercury in an area with little anthropic influence (except for the mercury source, which is supposedly long-range transport of presumably anthropogenic mercury). So where is the connection with sustainability (the focus of the journal)? In my perception, sustainability is the compatibility between human activities and a healthy environment. Apparently, the different degrees of grassland degradation are NOT connected with the presence of mercury, but are a consequence of other cause(s) – principally, salinization, if I understand correctly. So why putting throughout the paper all this emphasis on the different types of grassland? Mercury contents are fairly similar, which is consistent with a long-distance mercury source. The only remarkable difference is the daytime trend of atmospheric mercury, which deserves more discussion (my guess is that in the non-degraded grassland the vegetation cover delays the impact of solar radiation on soil – but it is just a guess). There is another point that definitely requires more discussion. At this stage, we don’t know which is the Hg concentration in your soil, because you report concentrations in ug/kg, which means nothing (see comment to § 3.1.1 below). If it is mg/kg, the content is quite high for a “normal” soil – in many countries, these soils would be considered seriously polluted; indeed you say that such a content exceeds Chinese standards and is much higher than other grasslands. So there must be an important mercury source (or more). You vaguely refer to “surrounding cities” – however no city is shown in Fig. 1, and more important there is no specific information on their role as mercury sources (size? Presence of industries?)
R7: Yes. The objective of this paper is to investigate the (quasi-natural) biogeochemical cycle of mercury in an area with little anthropogenic influence.
We mainly want to explore how the mercury cycling process of grassland will change after grassland degradation, because we believe that the sustainability of grassland will change with the increase of degradation degree, so the mercury changes in soil, atmosphere and plants will also change during the reverse succession process of grassland.
The change of the atmospheric mercury levels, we think there are two factors, one is the different degradation degree of grassland as the surface coverage, delay the effects of solar radiation on the soil mercury, second, we found that the atmospheric mercury is closely related to the temperature, we also found the same in the other articles, the temperature, the higher the content of mercury in the atmosphere, this is mainly due to the temperature force
As for the surrounding cities and mercury sources, we have explained in the paper that although there is no obvious surface mercury source nearby, we believe that under such natural conditions, the main source of mercury is atmospheric deposition, especially from the surrounding industrial cities (such as Changchun), and the urban mercury circulates with the atmosphere, thus settling on the grassland.
C8: Lines 25-27: as you write it here, there is a contradiction between single factor and accumulation indexes (which suggest a slightly polluted environment), and the potential risk index, pointing to a severe pollution. You should comment on reasons for such a discrepancy, and make clear your stand (do you believe that the area is slightly polluted, or severely polluted?). See further remarks on the confusing, in places contradictory, use of "light", "moderate" "severe"....
R8: Due to the different evaluation factors and methods used by different evaluation indicators, the evaluation results will be different. We mainly refer to the main results under different evaluation methods. According to the data of soil mercury content, we believe that the area is at a moderate pollution level and its ecological risk is very high.
C9: Line 64: plants roots adsorb [mercury] through transpiration? Is it so?
R9: This is our wrong expression, and we have changed it.
C10: Line 135: “geographical location”: I suppose you mean the local geomorphology
R10: Yes, Our means is “local geomorphology”. We have changed it.
C11: Line 139: “due to low disturbance degree”: unclear meaning – how can a low disturbance degree be responsible for the presence of mercury in soil?
R11: What we mean here is that the area is subject to low degree of external anthropogenic or industrial disturbance, and soil mercury mainly comes from atmospheric deposition and soil background value.
C12: Line 155: GB/T 37067-2018: what does this acronym mean? Please describe in full the criteria for assigning the degradation status
R12: We changed it. GB 19377-2003 mean China National Standard: Parameters for degradation, sandification and salification of rangelands. And we described the grassland degradation level in Table 2.
C13: Line 163: “heavy soil capacities”: cannot understand what does this mean
R13: Sorry, our mean is “soil bulk density is high”, and we changed it.
C14: Lines 166-167: why was moderately degraded grassland given priority?
R14: It is wrong, our mean is “the moderately degraded grassland community is dominated by miscellaneous grass community”, we changed it.
C15: Line 204: see general comment #3
R15: Yes, we made changes.
C16: Line 207: “after treatment”: what kind of treatment?
R16: We changed it. “after treatment” to “after sieving”.
C17: 3.1.1 and following – you report concentrations in ug/kg, which means nothing – is it mg (micrograms)? mg (milligrams)?
R17: Yes, our unit of measurement is ug/kg.
C18: Lines 276-278: indeed values for the non-degraded grassland have a much larger SD – any explanation for this? I suggest you provide also median values. As suggested in the previous review, a figure showing boxplots would be helpful
R18: Thanks for your advice. We added the boxplots.
C19: Lines 279-280: given the SD, the three averages are not really different
R19: We provide a boxplots to help you see the data intuitively.
C20: Table 2: is n = 48 for all three grassland types?
R20: No. There were 48 samples in any type of grassland.
C21: 3.1.2: please report the formulas for calculation of the indexes, and which is the difference between the various indexes. Please explicit the criteria for defining pollution “light”, “mild” etc
R21: We provided the grading criteria of different evaluation indicators in Table 4.
C22: Lines 292-298: most indexes correspond to “mild pollution”, however for the cumulative index of the severely degraded soil you speak of “light pollution” – which is the difference?
R22: They have the same meaning, but to avoid misunderstanding and to be consistent with what has been written above, we have changed the term “slightly pollution”.
C23: Lines 309-315: reasoning here is pretty confusing. You start saying that “the moderate degradation grassland had the highest potential risk index”; from what I can see in Fig. 4, the maximum value of Igeo is 141.2 for the moderately degraded soil, and 141.5 for the severely degraded soil. Then you say “the potential risk values of the soil with moderate degradation were lower than the moderate risk values” – which are these “moderate risk values”?
R23: Here are our errors of expression. “the potential risk values of the soil with moderate degradation were lower than the moderate risk values” changed to “the potential risk values of the soil with severe degradation were lower than the moderate risk values”.
C24: Lines 310 and 315: I suppose that the decrease of the potential risk with depth corresponds to a decrease in the mercury content. This would support the idea that the main source of mercury in the area is long-distance transport of airborne mercury. You may want to comment on this.
R24: Yes, the decrease of the potential risk with depth corresponds to a decrease in the mercury content.
C25: Fig. 4: one of the horizontal axes has no label (that with 2-3 marks); what does it represent?
Lines 328-334: I do not feel the differences in maximum values are really significant – which is the experimental uncertainty (see general comment #3….)? From what you show in Fig. 5, the maximum value over the severely degraded soil corresponds to a spike at about 14:00 (not noon as written in the text). The general trend is perfectly overlapping with the moderately degraded area, and the maximum spike could be just a transient event (such as a wind gust – typical in this kind of measurements…). What you can safely say is a) there are no significant differences among the three areas in term of absolute values; b) the non-degraded area has a quite distinct temporal trend (maximum at about 17:00 – not 16:00, at least from what I see in Fig. 5). Can you suggest a reason for this difference?
R25: 1,2, and 3 in the horizontal axis represent individual sample potential risk assessment index values. We have marked it in the annotations in Figure 4.
We agree with your modification proposal. We believe that the maximum mercury content occurs at around 14 p.m., the main reason is that Due to the large temperature difference between day and night in Northeast China, when the sun shines on the grassland, the surface temperature cannot rise rapidly, but there is a gradual temperature accumulation process. When the surface temperature reaches the highest, it should be around 14 p.m. at this time, the surface temperature is high, and the soil mercury element is easy to be reduced to mercury vapor at high temperature and volatilize into the atmosphere, Make the mercury content reach the maximum at this time.
C26: 3.2.2: all this section is quite confusing. You speak about 3D maps – where is the third dimension in Fig. 6? First you say (line 347-348) that the conditions are “under severe risk”, then you speak (lines 355-356) of “moderate risk” and “slightly polluted level” (once and again please explain clearly the criteria for defining “slight”, “moderate”, “severe”….). From Fig. 6, the patterns for the three grassland types look pretty much the same: how can you say that the moderately degraded grassland has the highest potential risk (line 350), and that “the time range of the highest risk value changed with the different grassland types” (line 360-361)? Lines 351 and following: How can the risk be reduced by an increase of the measuring time?
R26: We did not add the 3D diagram in this manuscript, so we revised this sentence.
We have rewritten 3.2.2
C27: 3.3, eqn (4): please give the statistical significance (p value)
R27: We have supplemented the p value.
C28: Lines 370-371: the correlation works in both ways: soil mercury can derive from adsorption of atmospheric mercury; conversely, atmospheric mercury may be re-emission from soil (as strongly suggested by the time trend: more irradiation, more mercury in the atmosphere)
R28: Yes, We have added.
C29: Line 377: graphs of Fig. 7 does not make easy to see differences (the figure should be complemented by the full set of numerical values, as repeatedly recommended). I can only see that in the non-degraded grassland the overall content of mercury in plants is lower than in the other two types, but other differences are not apparent
R29: Because of there is no significant difference in mercury content between plant parts of different types of grassland, the difference cannot be seen in figure 7.
C30: Lines 403-405: here do you refer to non-degraded grassland?
R30: Yes, We have added it.
C31: Lines 417-418: you say “the sources of the pollution could be located” – where and what? There is nothing in this respect
R31: We thought the sentence was misexpressed, so we removed it. Using this method does not predict the source of pollution.
C32: Lines 426-430: this information is more adequate for the Introduction
R32: Thanks for your advice. We have moved it to the Introduction.
C33: 4.1: this section has a weak logical structure. It starts with a comparison with other grassland areas (lines 430-433); it is a pure list with no discussion (reason(s) for the differences?). Then there are very general statements (lines 435-442) about mercury sources/transport: what is the connection with this paper? finally (lines 443-453) some speculations about the role of organic matter (you provide NO data on the organic content of soils). There is no obvious connection between the three parts.
R33: We rewrote the contents of this section. The main purpose of listing other grasslands is to refer to the pollution risk of mercury in different grasslands. For the problem of soil organic matter, we learned in previous studies that the content of soil organic matter will affect soil mercury. Therefore, when we explore the changes of soil mercury in different degraded grasslands, we speculate that it may affect soil mercury because there are significant differences in the content of organic matter on the surface of different degraded grasslands (these data are in our other submission papers). We have modified this part.
C34: Line 433: here you define the pollution “severe”, elsewhere you say “moderate”, “slight”…. Please be clear and consistent!
R34: What we're talking about here is the potential risk index rating as severe. We have defined our statements in the text.
C35: Line 470: there are no cities shown in Fig. 1. Please indicate the largest centers. Are there data on mercury emissions from these cities? Are they likely sources in terms of wind directions?
R35: We have added to this section. “The atmospheric mercury sources in that region may also be related to the several surrounding cities, such as Changchun, a typical industrial city in northeast China. This area may be affected by the spring wind direction (southeast wind) resulting in increased atmospheric mercury deposition.”
C36: Lines 471-479: quite general statements with little demonstrable connection to this paper
R36: Thank you for your advice, we have deleted this nonsense paragraph.
C37: Line 488: from Fig. 5 it seems that the highest mercury content occurs at about 14:00
R37: Yes. We have changed it.
C38: Lines 489-490: unclear what you mean with “positive effects of temperature on deposition of mercury”; do you mean there is a positive correlation?
R38: Yes. This is what has been shown in the studies that have been done.
C39: Lines 491-493: what has this to do with this paper? No data presented in this respect.
R39: As a result of our research involves in plant, so we discussed the possible mechanisms of mercury in the atmosphere circulation of plants.
C40: Lines 501-502: “This deposition occurs….”: unclear sentence. In principle, atmospheric deposition occurs on living plants and bare soil (it may also occur on a piece of dead wood, of course, but in general it is negligible). When leaves fall, or the entire organism dies, mercury is transferred to the soil. Overall, this sentence adds nothing, and can be removed
R40: Thanks for your advice. We have removed it.
C41: Lines 504-515: once and again, a sequence of general statements without specific connection to data presented in this paper
R41: Thanks for your advice. We have removed it.
C42: Lines 520-521: the average content in the non-degraded grassland is lower than in the other two types. Moreover, as pointed out before, differences are small and probably not significant.
R42: Yes, we changed it.
C43: Line 525: “as a result” of what? How can stomatal absorption result in variations? See also comment to Line 377.
R43: We believe that atmospheric mercury deposition may be derived from plant leaves, and we changed in text.
C44: Lines 533-536: oversimplification of rather complex processes for which you have little control. First, as previously said, mercury differences in the three types of soil can hardly be considered significant; second, the conclusion that “more vegetation coverage = less mercury in individual plants” implies that the limiting factor is the atmospheric concentration of mercury: maybe so, but it is just a hypothesis
R44: Yes. We infer that this is the case.
C45: Line 538: from Fig. 7, Hg contents in plants in the moderately degraded grassland look quite similar to the severely degraded grassland
R45: We have amended this sentence in the text.
C46: Lines 544-545: “suitable species” for what?
R46: “suitable species” means “plants with a high concentration of mercury”, we have changed it in text.
C47: Lines 545-546: L. chinensis is the only species you studied. You can say nothing about others.
R47: No. But Leymus chinensis is the dominant species in our study area, and they are abundant. Our main research object is Leymus chinensis.
C48: Lines 553-554: with “transformation” you mean entering the food chain?
R48: No. Our mean is “mercury transfer between soil and plant is an important part of mercury cycle in Songnen grassland.”
C49: Line 558: from Table 2, the largest difference is for non-degraded grassland.
R49: In Table 3, the average value of soil mercury content is the highest in moderately degraded grassland.
C50: 5 (Conclusions): reiterates many questionable statements addressed in the previous comments. This section should be rewritten after addressing the above points.
R50: Thanks for your advice. This section have be rewritten.
C51: Lines 558-559: “vegetation degradation may affect mercury cycling”: there is nothing in this paper that supports such a conclusion
R51: For meaningless words, we have deleted them.
C52: Lines 559-560: from Table 2, the lowest average Hg content is for non-degraded soil
R52: Yes, we changed it.
C53: Lines 561-563: from Fig. 4, the lowest risk appears associated with severely degraded soil; the other two types do not show appreciable differences
R53: Thanks for your advice, we have changed it.
C54: Lines 564 and following: light pollution, low risk, moderate risk… please define clearly!
R54: We have redefined them.

Reviewer 2 Report
I do not have comments.
Author Response
Dear reviewer:
Thank you very much for your support of our paper.
We have modified other problems in the article. We highlighted all the changes by using a red font in the revised manuscript.
Thank you very much!
Yours sincerely,
Gang Zhang behalf of the authors.

Round 3
Reviewer 1 Report
While I appreciate the authors' efforts to improve the paper, I must notice that their revisions always sound done in much haste and somehow carelessly. This is shown for instance by the many typos plaguing the text (e.g., china instead of China). Once and again I made annotations in the attached pdf (some are repetitions from previous rounds of reviews, and were clearly overlooked). Several points remain open; I concede that some may be a subjective matter. There are, however, a couple of issues that must be solved
- You keep using ug/kg as a concentration unit: such a unit DOES NOT EXIST - it is either µg/kg or mg/kg
- I cannot understand your refusal to provide the full data set - you have no sound reason to do so. Release of full data is a very useful contribution to readers. I respectfully point out that it is not a matter of what you think: it is a basic principle of open science and an explicit requirement of the journal.
- A minor point: please show the position of Changchun city (from Google maps I appreciate that it lies southeast of the study area, right?)

Author Response
From: Gang Zhang
School of Environment, Northeast Normal University,
Key Laboratory of Vegetation Ecology, Ministry of Education, Northeast Normal University,
Institute of Grassland Science, Northeast Normal University,
Changchun Jilin, 130117, China.
September 23 2021
Dear Reviewer
Thank you for considering the revised version of our manuscript “Characteristics of Mercury Pollution and Ecological Risk Assessment in Different Degraded Grasslands of the Songnen Plains, Northeastern China (Ms. ID.: sustainability-1339792)”, for publication in Sustainability. We appreciate all the comments from the reviewer on the previous manuscript. We have thoughtfully considered these comments. The following are the itemized responses to the reviewers’ comments (given in blue below). We highlighted all the changes by using a red font in the revised manuscript.
Thank you very much!
Yours sincerely,
Gang Zhang behalf of the authors.
